# Basketball robot object detection and distance measurement based on ROS and IBN-YOLOv5s algorithms

Jirong Zeng *, Jingjing Fu

Gannan Normal University, Ganzhou, China

* GNSYzjr@163.com

**Data Availability Statement:** All relevant data are within the manuscript.

**Funding:** The author(s) received no specific funding for this work.

## Abstract

With the combination of artificial intelligence and robotics technology, more and more professional robots are entering the public eye. Basketball robot competition, as a very good target system for autonomous robot research, is very suitable for conducting research on robot autonomous perception system object detection. However, traditional basketball robots have problems such as recognition difficulties, which seriously affect the recognition of robot targets and distance measurement based on recognition. To improve the performance of basketball robots in competitions, research was conducted to improve the object detection system. Firstly, a basketball robot object detection system based on robot operating system was designed. In the software layer of the object detection system, an algorithm that combines YOLOv5s and laser detection was used, and an appropriate instance batch normalization network module was introduced in the YOLOv5s algorithm to improve the model's generalization ability. The experiment outcomes indicated that the improved algorithm had intersection over union (IoU), structural information loss, ambiguity and signal-to-noise ratio of 0.96, 0.03, 0.13, and 0.98, respectively, and performed the best in the other comparison models. The recall curve area and F1 value of the improved algorithm were 0.95 and 0.9789, respectively. In the detection of basketball, volleyball, and calibration columns, the average classification accuracy of the improved model was 95.87%, and the average calibration box accuracy was 97.05%. From this, the algorithm proposed in the study has robust performance and can efficiently achieve object detection and recognition of basketball robots. The improved algorithm proposed in the study provides more reliable and rich information for the perception ability of basketball robots, as well as for their subsequent decision-making and action planning, thereby improving the overall technical level of the robots.

## 1. Introduction

In response to the call for a strong sports and technology country, various robot competitions have emerged in various associations and schools. At present, basketball robots have been widely used in various robot competitions [1]. In the Chinese Robot Competition, the

**Competing interests:** The authors have declared that no competing interests exist.

basketball robot project has become an important event [2]. Due to the fact that participating basketball robots must locate balls of specific colors on the field and complete a series of passing and pitching tasks to earn points, a highly accurate object detection system is crucial for the precise positioning of basketball robots [3]. When studying the operation system of basketball robots, it is necessary to clarify the position and posture information of the basketball robot, so the design of object detection algorithms plays a crucial role. Object detection algorithm is a critical technology in the field of computer vision, which can recognize and locate specific objects in images or videos. The object is to accurately find the bounding box of the object in the image and classify it, thereby achieving effective recognition of the object [4].

Presently, conventional object detection algorithms, including binocular vision algorithms, deep convolutional neural networks, support vector machines, and others, despite exhibiting high recognition accuracy, are characterized by a slow detection speed, poor learning abilities, and limited generalization capabilities [5]. In the context of basketball robot games, the object detection system equipped not only requires high detection accuracy and fast calculation, but also strong feature learning ability to cope with the impact of different lighting environments on the playing field. Therefore, based on robot operating systems, this study introduces the lightweight and high detection accuracy You Only Look Once version 5 small (YOLOv5s) algorithm as the object detection algorithm, and integrates an appropriate amount of instance-batch normalization network modules to improve the original algorithm and enhance its detection performance.

The research structure is divided into four parts. The section 1 is a review of relevant research. Section 2 is the design of a basketball robot object detection system based on Robot Operating System (ROS) and improved YOLOv5s. Section 3 is an experimental analysis of the performance of the object detection system. Section 4 is a research summary and shortcomings, and proposes future research directions. ROS, as a flexible robot software framework, can effectively manage and coordinate robot hardware and software components, thereby improving the overall performance and efficiency of robots. As an improved object detection algorithm, the research algorithm enhances the generalization ability of the model by introducing the Instance and Batch Normalization Networking (IBN-Net) module, and improves the detection accuracy when there is a large difference in the distribution between the test set and the training set. This is crucial for robots to adapt to object detection tasks in different environments and conditions.

## 2. Related works

ROS is currently the most widely used robot software development platform, which has been extensively studied by scholars due to its rich toolset, distributed design, and excellent scalability. Tosun et al. proposed a hybrid algorithm to address the path planning issue of quadcopter drones in dynamic and static environments, and combined this model with ROSs to conduct performance benchmark tests in different environments. The research results showed that the model had significant effects in both planning and time [6]. Guo used an improved YOLOv5 model to detect various road surface diseases, optimized the YOLOv5 model, and introduced an attention mechanism to enhance the robustness of the model. The improved model was more suitable for deployment in embedded devices. The experimental results showed that compared with YOLOv5s and YOLOv4 models, the improved YOLOv5 model has increased mean Average Precision (mAP) by 4.3% and 25.8%, respectively. This method could provide technical reference for road disease detection robots [7]. Singh et al. developed an obstacle avoidance algorithm for four-wheel mobile robots. Firstly, a dynamic model was established using key and graph methods. Then, a navigation model for the four-wheel mobile robot was

established using ROS. Finally, a prototype model of the four-wheel mobile robot was established. In simulation testing, the effectiveness of the model was demonstrated by testing single and two position obstacles in the mapping environment, which could effectively achieve safe driving [8]. Oliveira et al. proposed a vehicle data acquisition and analysis surface model with the middleware framework ROS system. This model had devices that could achieve synchronous audio and video acquisition and storage, combined with facial detection algorithms in the data collected from cameras and microphones. The evaluation was carried out through the implementation of an inference system. The experiment findings indicated that the model could track abnormal situations by monitoring the actions of drivers and passengers [9].

YOLOv5s is an object detection network with anchor boxes. Due to the lightweight implementation of the network structure, it can perform well in both detection speed and accuracy, and is widely used in many fields. The apple target recognition algorithm is one of the core technologies of apple picking robots. To automatically identify the apples that can be grasped and those that cannot be grasped in apple tree images, Bin proposed a lightweight apple target detection method for picking robots using an improved YOLOv5s. The experimental results showed that the proposed improved network model had recognition recall, accuracy, mAP, and F1 value of 91.48%, 83.83%, 86.75%, and 87.49%, respectively. The mAP of the improved YOLOv5 model increased by 5.05%, 14.95%, 4.74%, and 6.75%, respectively, and the model size was compressed by 9.29%, 94.6%, 94.8%, and 15.3% [10]. Zhang et al. proposed an automatic detection method based on YOLOv5s to solve the issue of silicon detection on wheat straw epidermis. The input was modified by adding a reverse residual unit (Resunit), point by point convolution, and attention mechanism, while cutting off the focus module. The experiment findings indicated that the model could be transplanted to mobile devices, and the accuracy of silicon detection on wheat straw was as high as 98.88% [11]. Liu et al. constructed a recognition and positioning system with improved YOLOv5s and depth camera for chili picking and positioning. The study combined the improved algorithm with a bidirectional feature pyramid network deep learning model, aiming to effectively detect the deep features and high-precision of chili peppers. The experiment findings indicated that the model had good robustness in day and night, as well as in leaf occlusion scenes [12]. Qian et al. proposed a method of first segmentation and then detection for hot spot defect detection in photovoltaic modules. The improved semantic segmentation model was used to extract the photovoltaic module area, and the object detection model derived from YOLOv5s was used to detect hot spot defects from the segmentation area. The experiment findings indicated that the average intersection of the photovoltaic module semantic segmentation model was 97.8%, and the average accuracy was 89.6% [13]. Targeted spray has always been one of the hot issues in the research field of plant protection robots. To solve the problem of precise spraying, Liu proposed a soybean phenotype information perception method based on improved YOOv5. The experimental results showed that the improved YOLOv5 had an mAP of 96.13%, FPS of 79, a 39% reduction in the number of model parameters, and a 55.56% reduction in weights. The correlation coefficient between the obtained data and the manual measurement of plant phenotype information was 0.98384, indicating higher accuracy and better consistency [14].

From this, although there are many research results on ROS and YOLOv5s, most of them have not conducted in-depth research on problems such as low generalization ability, poor learning ability, complex network training, and difficulty in convergence during detection. Therefore, the study introduces IBN-Net sections that can improve learning and generalization abilities into the YOLOv5s object detection algorithm, and build an ROS basketball robot object detection system based on this improved model. Comparative tests were conducted on the given dataset to prove the specific performance of the model. **Summary table of related work as shown in Table 1**.

**Table 1. Summary table of related work.**

| Author | Time | Data set | Method | Performance indicators | Result |
|---|---|---|---|---|---|
| Tosun et al. | 2023 | Dynamic and static environments | Combination of Hybrid Algorithm and ROS | Planning efficiency, time efficiency | Has a significant impact on both planning and timing |
| Guo | 2022 | Road surface disease image | Improved YOLOv5 model with introduction of attention mechanism | mAP | Compared to YOLOv5s and YOLOv4, mAP increased by 4.3% and 25.8%, respectively |
| Singh et al. | 2021 | Simulate map environment | Building dynamic models using key and graph methods, ROS navigation model | Success rate of obstacle avoidance | Effectively achieving safe driving in simulation testing |
| Oliveira et al. | 2022 | Driver and passenger behavior data | ROS system vehicle data collection and analysis model, combined with facial detection algorithm | Abnormal behavior monitoring capability | Track abnormal situations by monitoring driver and passenger behavior |
| Bin et al. | 2021 | Apple Tree Image | Improved YOLOv5s Lightweight Apple Object Detection Method | Identify recall, accuracy, mAP, F1 value | The recall rate is 91.48% and the accuracy rate is 83.83%, mAP 86.75%,F1 value 87.49% |
| Zhang et al. | 2023 | Silicon image of wheat straw epidermis | Automatic detection method based on YOLOv5s, adding reverse Resunit and attention mechanism | Accuracy rate | The accuracy rate is as high as 98.88% |
| Liu et al. | 2023 | Chili image | Improved YOLOv5s combined with bidirectional feature pyramid network | Robustness | Good robustness in scenarios of day, night, and tree cover |
| Qian et al. | 2023 | Photovoltaic module image | The method of segmentation before detection, using an improved semantic segmentation model and YOLOv5s | Average IoU, average accuracy | Semantic segmentation model IoU 97.8%, accuracy 89.6% |
| Liu et al. | 2023 | Soybean phenotype information image | A soybean phenotype information perception method based on improved YOLOv5 | mAP, FPS, number and weight of model parameters | mAP 96.13%, FPS 79, 39% reduction in model parameters and 55.56% reduction in weights |

## 3. Design of basketball robot object detection software system based on ROS and IBN-YOLOv5s

The first section introduces the basic composition of ROS software, which uses a object detection algorithm combining laser detection and YOLOv5 in its application layer. The second section addresses the problem of poor learning and generalization ability of the YOLOv5s algorithm, and introduces the IBN-Net module to improve its performance.

### 3.1 Basketball robot object detection software system based on ROS

ROS consists of three modules: application layer, operating system layer, and hardware layer. The system treats the software functions of the robot as nodes, and each node can send messages to communicate with each other [14]. The node manager in the network communication mechanism manages and schedules the communication process between nodes, while providing a service for configuring global parameters. Combined with Ubuntu, an object detection system is designed, as denoted in Fig 1.

In Fig 1, the object detection algorithm that combines laser detection and YOLOv5 is selected as the application layer, while the operating system layer selects the Ubuntu system. At the hardware level, lasers and cameras are important hardware for object detection systems. The camera provides image data sources to YOLOv5. After successfully detecting the object, the laser provides distance and angle information of the object, and finally, the fused object calibration algorithm is used to measure the angle and distance of the object. During laser ranging, a light pulse is sent back after touching an object. The distance between the pulse and the object is calculated using Eq (1) [15,16].

$$d = \frac{1}{2}ct \tag{1}$$

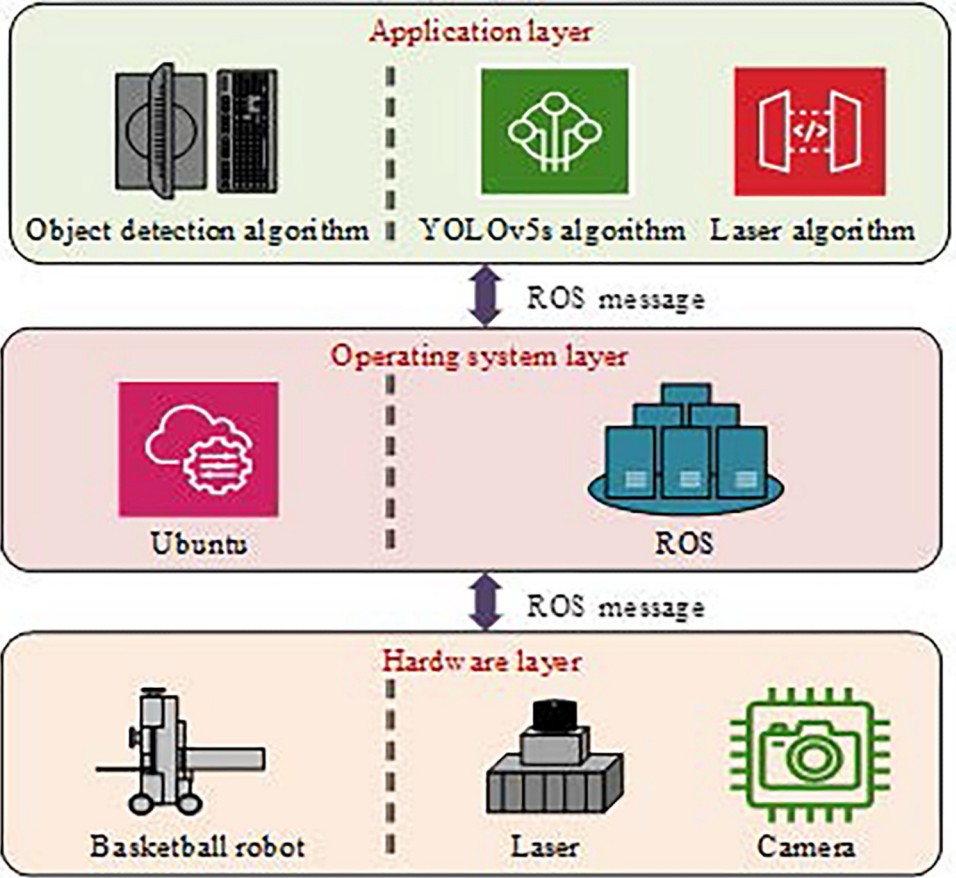

**Fig 1. Object detection software system for basketball robots.**

In Eq (1), *d*, *c*, and *t* respectively represent the measured laser distance information, light speed, and the total time from laser emission to return. The basketball robot game field and laser ranging scanning image are shown in Fig 2.

Fig 2A is a map of the basketball robot game field, and Fig 2B is a schematic diagram of laser detection. In Fig 2A, the basketball robot game field is 11 meters long and 7.5 meters wide, with only robots, object balls, and calibration columns in the field. Normally, robots perform lateral displacement detection within 1 meter of the object ball placement area and recognize at a distance of approximately 2–3 meters from the calibration column. The detection angle of the laser is [-45˚, +225˚]. Initially, the laser detection will cycle through the distance information obtained from each light wave. To prevent external objects from affecting the laser detection, all distances greater than 4 meters are processed as 4 meters. The calculation of laser detection distance and angle is shown in Eq (2).

$$
\begin{cases}
object\_distance = data\left[\dfrac{start + end}{2}\right] \\[2mm]
object\_angle = \left(\dfrac{start + end}{2}\right) \times 0.25° - 45°
\end{cases}
\tag{2}
$$

In Eq (2), *distance* means the distance. If there is a sudden change in the distance value

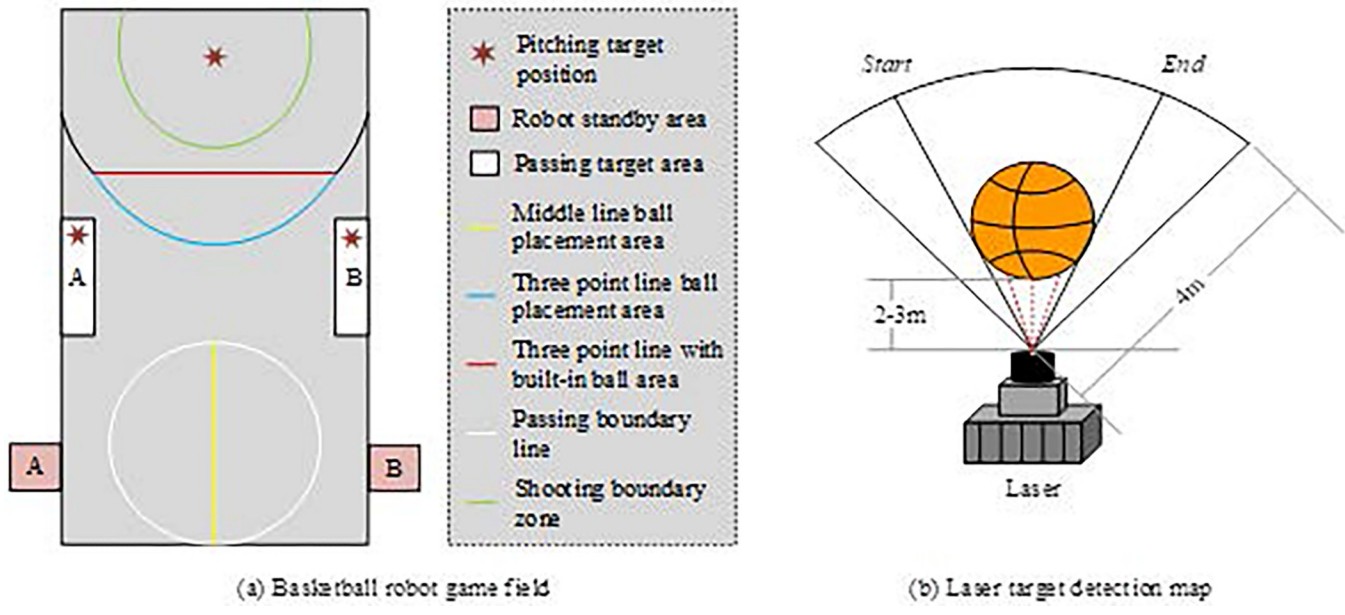

**Fig 2. Basketball robot game field and laser object detection map.**

between the $i$th and $i$+1th light waves and the difference is greater than the threshold, it is possible that the object has been detected, then the variable *start* = $i$+1. If the remaining distance values between the $n$ and $n$+1 light waves change again and the difference with the previous light wave is greater than the threshold, then it indicates that there is no detected object. At this time, the variable *end* = $n$. If the object is not detected, it will repeat the above steps until the object is found. YOLOv5 is an algorithm that combines detection speed and accuracy in the field of object detection. While meeting the requirements of basketball robot detection accuracy, its lighter model is more suitable for application in basketball robot controllers with higher real-time requirements compared to other algorithms in the YOLO series. The YOLOv5 model abstracts object detection in images as a regression problem. Considering that the laser detection algorithm can only detect the object and cannot determine what the object is, YOLOv5 is introduced in the study for object recognition. If the YOLOv5 algorithm determines that both detections are the same object, the laser is used to represent the angle distance of the object. The calculation rules for object angle are shown in Fig 3.

As shown in Fig 3, the robot is regarded as a coordinate system composed of $x$ and $y$ axes, the image is a coordinate system composed of $u$ and $v$, and the pixel center point coordinates are $(u_0, v_0)$. When determining the detection object, it first determines whether there is a object on the basketball court based on specific nodes. If there is, it will select the object closest to the basketball robot and with the largest volume for detection. The calculation method for area is shown in Eq (3).

$$object\_area = (x_{max} - x_{min}) \times (y_{max} - y_{min}) \qquad (3)$$

In Eq (3), $x_{max}$ and $x_{min}$ represent the max and mini values of the object on the $x$ axis, while $x$ and $y_{min}$ mean the max and mini values of the object on the $y$ axis. Secondly, it calculates the center point coordinates of the object with the largest area according to Eq (3), as shown in Eq

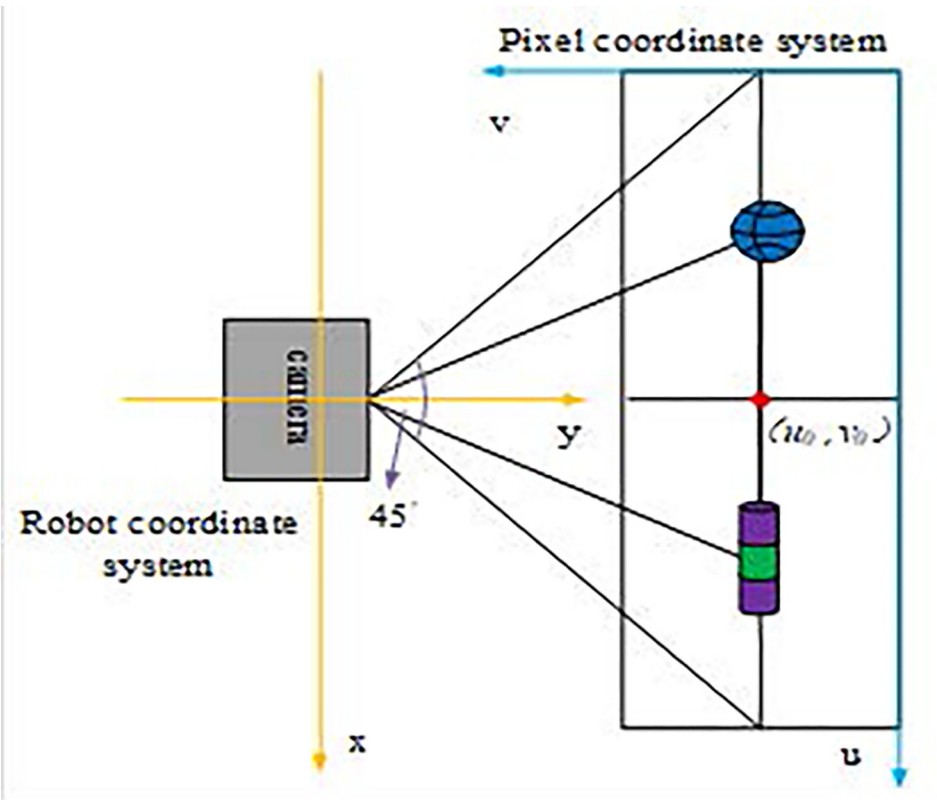

**Fig 3. Diagram for calculating the angle of the object sphere.**

(4).

$$\begin{cases} target\_bbox\_centerX = 0.5 \times (target\_bbox.x_{\max} + target\_bbox.x_{\min}) \\ target\_bbox\_centerY = 0.5 \times (target\_bbox.y_{\max} + target\_bbox.y_{\min}) \end{cases} \tag{4}$$

In Eq (4), *target_bbox* represents the object with the largest area in the object, *target_bbox* means the abscissa of the center point, and *target_bbox* means the ordinate of the center point. The angle calculation of the blue object located above the pixel center point $(u_0, v_0)$ in Fig 3 is shown in Eq (5).

$$\begin{aligned} target\_angle &= 90° + (u_0 - target\_bbox\_centerX) * d_u \\ &= 90° + (320 - target\_bbox\_centerX) * (45°/640) \end{aligned} \tag{5}$$

In Eq (5), $d_u$ represents the angle occupied by the pixel unit, and the $(u_0, v_0)$ coordinate of the pixel center point is (320,240). Since the camera perspective is, the angle occupied by each pixel unit is $d_u = 45°/640$. Similarly, it can be inferred that the angle of the cylindrical object located below the pixel center point is calculated using Eq (6).

$$\begin{aligned} target\_angle &= 90° - (target\_bbox\_centerX - u_0) * d_u \\ &= 90° - (target\_bbox\_centerX - 320) * (45°/640) \end{aligned} \tag{6}$$

In Eq (6), *object_angle* represents the angle of the object calculated using the laser, and

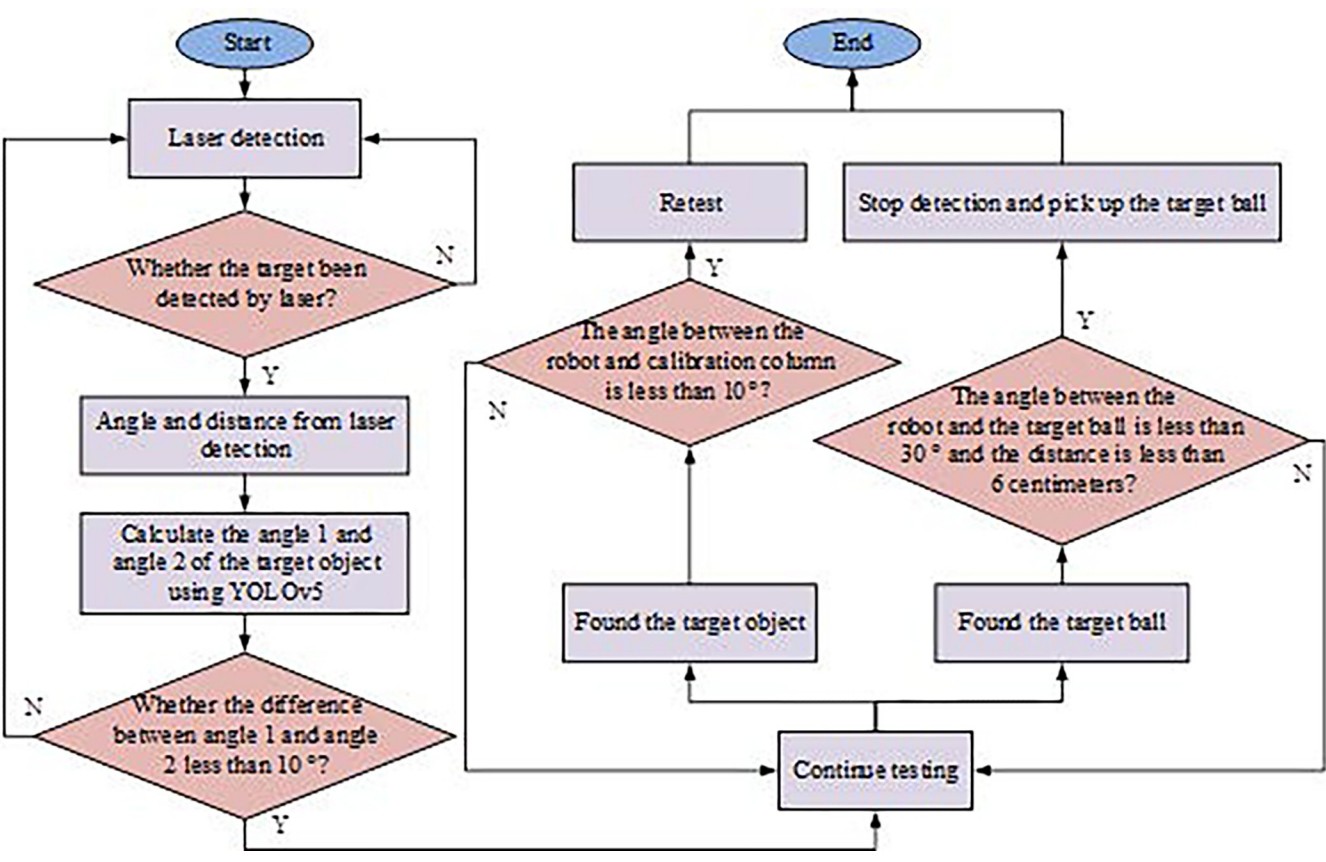

**Fig 4. Detection process of object detection system.**

*taget_angle* represents the angle of the object calculated by YOLOv5 under the camera. When the absolute difference between *object_angle* and *target_angle* is less than the set threshold, the detection object becomes the same, and the angle and distance of the object become *object_angle* and *object_distance*, respectively. After detecting the angle of the object through YOLOv5s, the system will combine the laser detection results with the data obtained from YOLOv5s for analysis. The workflow of the object detection system obtained through the above calculation steps is shown in Fig 4.

As shown in Fig 4, the object is first detected by laser. When the laser determines that there is no object, the detection is repeated. Otherwise, the angle and distance of the detected object are obtained. Then, based on the camera perspective, YOLOv5 is used to calculate the angle of the object. When the absolute difference between the two angles is less than 10°, it is determined that the object is to be found by both the laser and YOLOv5 algorithms. Subsequently, once it is confirmed that the detected object is a sphere, the robot will gradually move towards the sphere and pick it up. During this process, the object detection system will continuously use the object detection algorithm to update and provide feedback on the specific position and angle information of the sphere. When the distance between the robot and the object ball is less than 30 centimeters, due to the small distance, the image of the ball in the camera cannot be displayed as a whole. At this time, laser is used alone for detection. When the detection object ball is within 6 centimeters and the angle formed with the ball is less than 30°, the robot picks up the object ball and performs a throwing action.

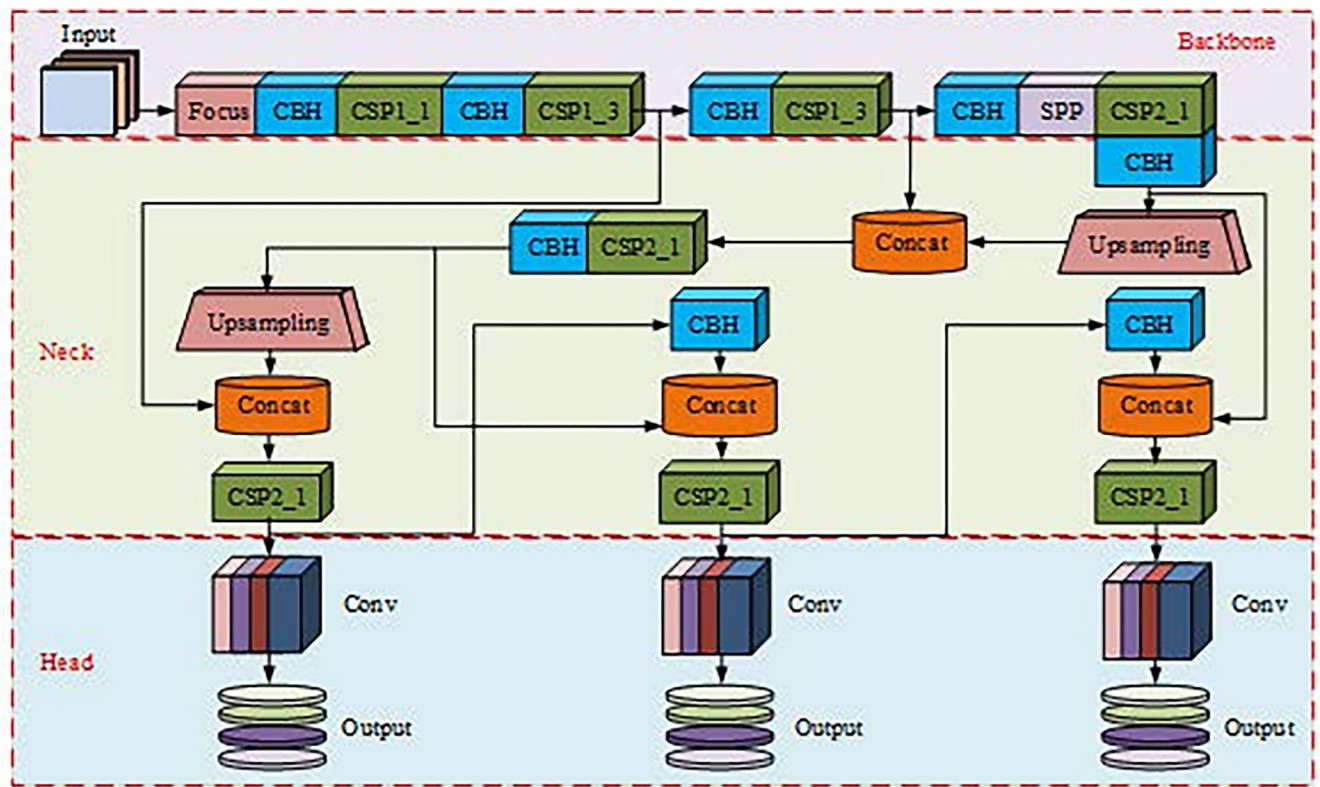

**Fig 5. Structure of YOLOv5s.**

### 3.2 Object detection design based on improved IBN-YOLOv5s algorithm

As the advancement of science and technology, basketball shooting robots have become the main project in robot competitions, and optimizing the object recognition algorithm of shooting robots has also become a current research hotspot. YOLOv5 can accurately detect multiple objects in an image and provide bounding boxes and category labels for the monitored objects. The object detection algorithm YOLOv5s selected for the study is the smallest model in the YOLOv5 model, which can effectively reduce the energy consumption of basketball robot movement. The YOLOv5s model consists of three modules: Backbone, Neck, and Head, and its structure is denoted in Fig 5 [17].

As shown in Fig 5, in YOLOv5s, Backbone includes four modules: Conv-Batch normalization HardSwish (CBH), Focus, Cross stage partial (CSP), and Spatial pyramid pooling network (SPP). CBH consists of convolutional layers, Batch Normalization (BN) layers, and HardSwish activation functions, while the CSP module adds an additional Resunit. Firstly, it inputs the sample into the Focus module, which divides the data information into four parts and generates a dimension channel. Next, the partitioned information is input into CSP for processing, and then into the Path Aggregation Network (PanNet), which includes several connection layers, convolutional layers, and CSP layers. However, due to the unpredictability of the basketball robot's playing field and the inability to control variables such as lighting, the YOLOv5s proposed earlier has an increased probability of incorrect recognition of objects. Therefore, for the scenario of basketball robots, BN and Instance Normalization (IN) modules are introduced in the study, and the instance and batch normalization networking you only look once version 5 small (IBN-YOLOv5s) model is proposed to raise the learning and generalization ability of

the model. BN is used to normalize the data distribution of each layer after convolutional layers and before nonlinear activation functions, effectively avoiding overfitting of the model and having a certain degree of regularization effect. Its expression can be found in Eq (7) [18].

$$BN(x) = \gamma \left( \frac{x - \mu_C(x)}{\sigma_C(x)} \right) + \beta \qquad (7)$$

In Eq (7), $\gamma$ and $\beta$ both represent the two affine parameters learned during training. The purpose of setting these parameters is to uniformly distribute all data in the model. $\mu_C(x)$ represents the mean of image calculation. $\sigma_C(c)$ represents the standard deviation of image calculation. Their definitions are shown in Eq (8).

$$\begin{cases} \mu_C(x) = \dfrac{1}{NHW} \sum\limits_{n=1}^{N}\sum\limits_{h=1}^{H}\sum\limits_{w=1}^{W} x_{nchw} \\ \sigma_C(x) = \sqrt{\dfrac{1}{NHW} \sum\limits_{n=1}^{N}\sum\limits_{h=1}^{H}\sum\limits_{w=1}^{W}(x_{nchw} - \mu_C(x))^2 + \varepsilon} \end{cases} \qquad (8)$$

In Eq (8), $x$ represents the input image tensor, $n$ means the number of samples, $W$ means the width of the image, $H$ means the height of the image, $N$ represents the number of samples in a training batch, $C$ means the color channel, $C$ represents a constant, and $\mu_C(x)$ in Eq (7) is not equal to 0. Therefore, $\varepsilon$ takes $10^{-5}$. $x_{nchw}$ represents the pixel value of the input tensor $x$, while $n$, $c$, $h$, and $w$ represent the $n$th sample, $c$th channel, height value $x$, and width value $w$, respectively. The principle of IN is similar to BN, which normalizes the channels of a single sample. Style normalization can be achieved by normalizing the mean and variance. Its calculation is shown in Eq (9).

$$IN(x) = \gamma \left( \frac{x - \mu_{nC}(x)}{\sigma_{nC}(x)} \right) + \beta \qquad (9)$$

In Eq (9), the affine parameters $w$ and $\beta$ generate images with different styles by changing the size of the parameter values. The definitions of $\mu_{nC}(x)$ and $\sigma_{nC}(x)$ are shown in Eq (10).

$$\begin{cases} \mu_{nC}(x) = \dfrac{1}{HW} \sum\limits_{h=1}^{H}\sum\limits_{w=1}^{W} x_{nchw} \\ \sigma_{nC}(x) = \sqrt{\dfrac{1}{HW} \sum\limits_{h=1}^{H}\sum\limits_{w=1}^{W}(x_{nchw} - \mu_{nC}(x))^2 + \varepsilon} \end{cases} \qquad (10)$$

In Eq (10), $\mu_{nC}(x)$ and $\sigma_{nC}(x)$ respectively represent the mean and standard deviation of each feature channel for a single instance image, and the meanings of other parameters are consistent with those in Eq (8). IBN-Net combines BN and IN in an appropriate form to work together on a dataset. IBN-Net combines BN and IN together, enabling them to work together on the dataset. IN learns features in images that are not affected by appearance changes, thereby improving the model's generalization ability. However, BN effectively preserves content related to images, which helps improve training speed and learning ability. IBN-Net combines the advantages of both, eliminating appearance differences while preserving the resolution of image content. Therefore, the study introduces IBN-Net into the YOLOv5s model and proposes an improved network model of IBN-YOLOv5s, as shown in Fig 6.

In Fig 6, Fig 6A showcases the original Resunit structure. The study created a new CIBH module by replacing some BN channels in the residual path with IN, as shown in Fig 6B. The modified Resunit is shown in Fig 6C and 6d. After adding the IN module, the module

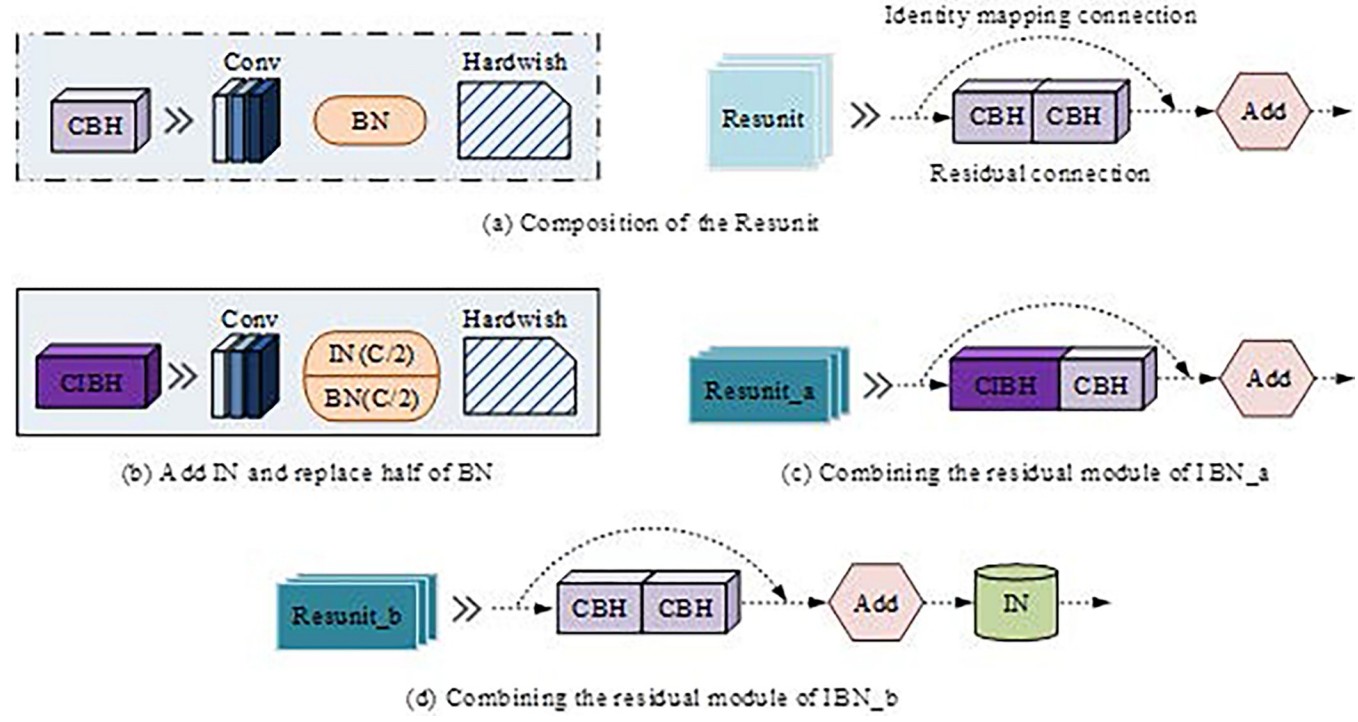

**Fig 6. Improved network model of IBN-YOLOv5s.**

performs the addition operation of residual paths and identity mapping paths. It modifies the Resunit with appropriate number of layers in YOLOv5s to Resunit_a and Resunit_b. As most of the appearance differences exist on the surface, the study only introduces appropriate IN modules into shallow structures in the Backbone module. Excessive introduction of IN will actually reduce model accuracy.

## 4. Experimental analysis of a basketball robot object detection system based on ROS and IBN-YOLOv5s

To prove the feasibility of the improved IBN-YOLOv5s algorithm in basketball robot object detection, the first section of the study conducted comparative experiments between IBN-YOLOv5s and other different object detection algorithms. The second section conducted experimental tests on the basketball robot object detection system based on the IBN-YOLOv5s object detection algorithm.

### 4.1 Comparative testing of IBN-YOLOv5s improved algorithm

In the experimental deployment, a desktop computer with 16 GB of RAM running on Windows 10 operating system was studied, and the CPU model was equipped with Intel (R) Core (TM) i5-9400F CPU. The GPU model was NVIDIA RTX 3080. In the training of deep learning libraries, PyTorch was selected for research. It is easy to learn that it has a large number of high-quality pre-trained models and open source projects for reference and use, and has a wide range of applications in the field of object detection. It supports the implementation of various object detection algorithms, including the YOLO series. Its high-performance computing power can meet the real-time requirements of basketball robot games. The dataset used was a self-built dataset of 3000 basketball game images, which included clear, blurry images

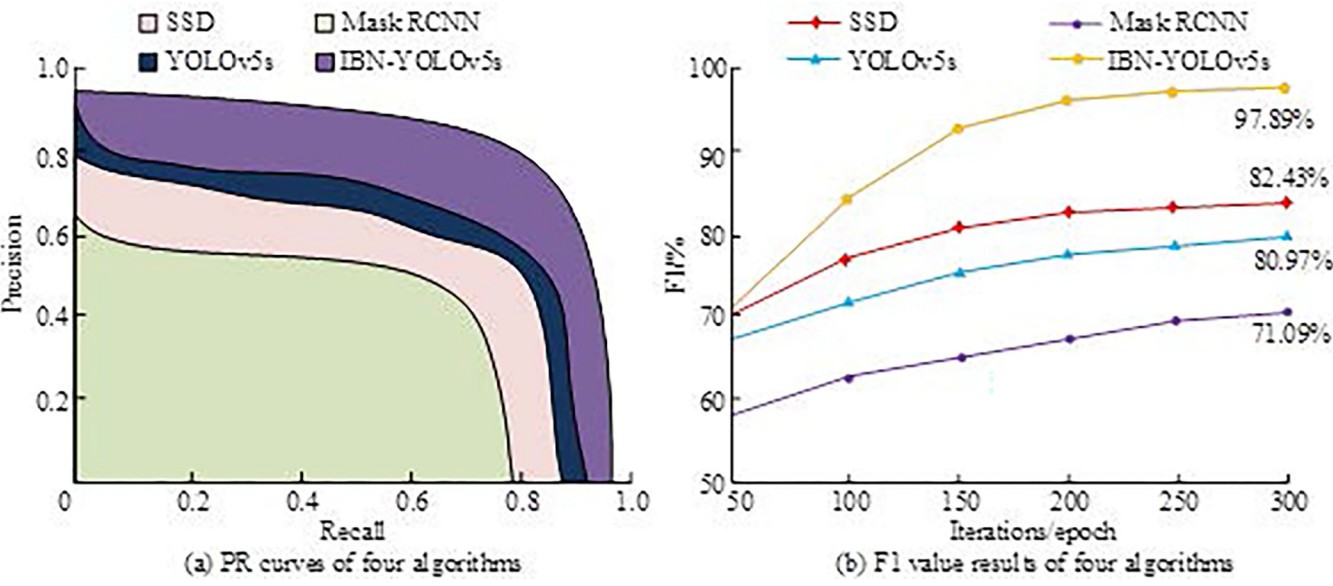

**Fig 7. PR curve and F1 value results of four algorithms.**

with different levels, and images with different lighting intensities. The dataset was broken into training and testing sets in a 7: 3 ratio. The study selected Single Shot MultiBox Detector (SSD), Mask Region-based Convolutional Neural Network (Mask RCNN), YOLOv5s, and the improved IBN-YOLOv5s algorithm proposed by the research for comparative testing. Firstly, the performance on the Precision Recall (PR) curve and F1 value is shown in Fig 7. The maximum number of iterations for the experimental setup was 300. The maximum number of iterations for model training was 400.

Fig 7 showcases the PR curves and F1 values of four algorithms in the training set. Fig 7A showcases the PR curve. The areas enclosed by the SSD, Mask RCNN, YOLOv5s, and IBN-YOLOv5s curves and the coordinate axis were 0.82, 0.69, 0.87, and 0.95, respectively. IBN-YOLOv5s had the largest curve area and the highest detection precision. Fig 7B shows the F1 value variation curve, and as the amount of iterations increased, the F1 values of all algorithms improved. When the number of iterations was 50, the F1 values of SSD, Mask RCNN, YOLOv5s, and IBN-YOLOv5s were 64.23%, 59.59%, 68.21%, and 71.56%, respectively. When the number of iterations was 150, the F1 values of SSD, Mask RCNN, YOLOv5s, and IBN-YO-LOv5s were 82.43%, 71.09%, 80.97%, and 97.89%, respectively. The IBN-YOLOv5s model maintained the highest F1 value before and after iterations. When the amount of iterations exceeded 150, the F1 value changed gradually and stabilized at over 90%. The comparison of intersection to IoU, structural information loss (SIL), ambiguity (BM), and signal-to-noise ratio (SNR) of images processed by various algorithms is shown in Fig 8.

Fig 8A showcases the IoU of the processed image, Fig 8B represents the SIL of the processed image, Fig 8C showcases the comparison of BM of the processed image, and Fig 8D denotes the comparison of SNR of the processed image. In Fig 8A, as the number of training sets increased, the IoU values of the processed images all showed an increasing trend. When the training set size was 800, the IoU values of SSD, Mask RCNN, YOLOv5s, and IBN-YOLOv5s were 0.76, 0.89, 0.85, and 0.96, respectively. In Fig 8B, as the size of the training set increased, the SIL values of each algorithm gradually decreased. When the size of the training set was 800, the SIL values of SSD, Mask RCNN, YOLOv5s, and IBN-YOLOv5s were 0.24, 0.12, 0.19, and 0.03, respectively. In Fig 8B, as the size of the training set increased, the image processing

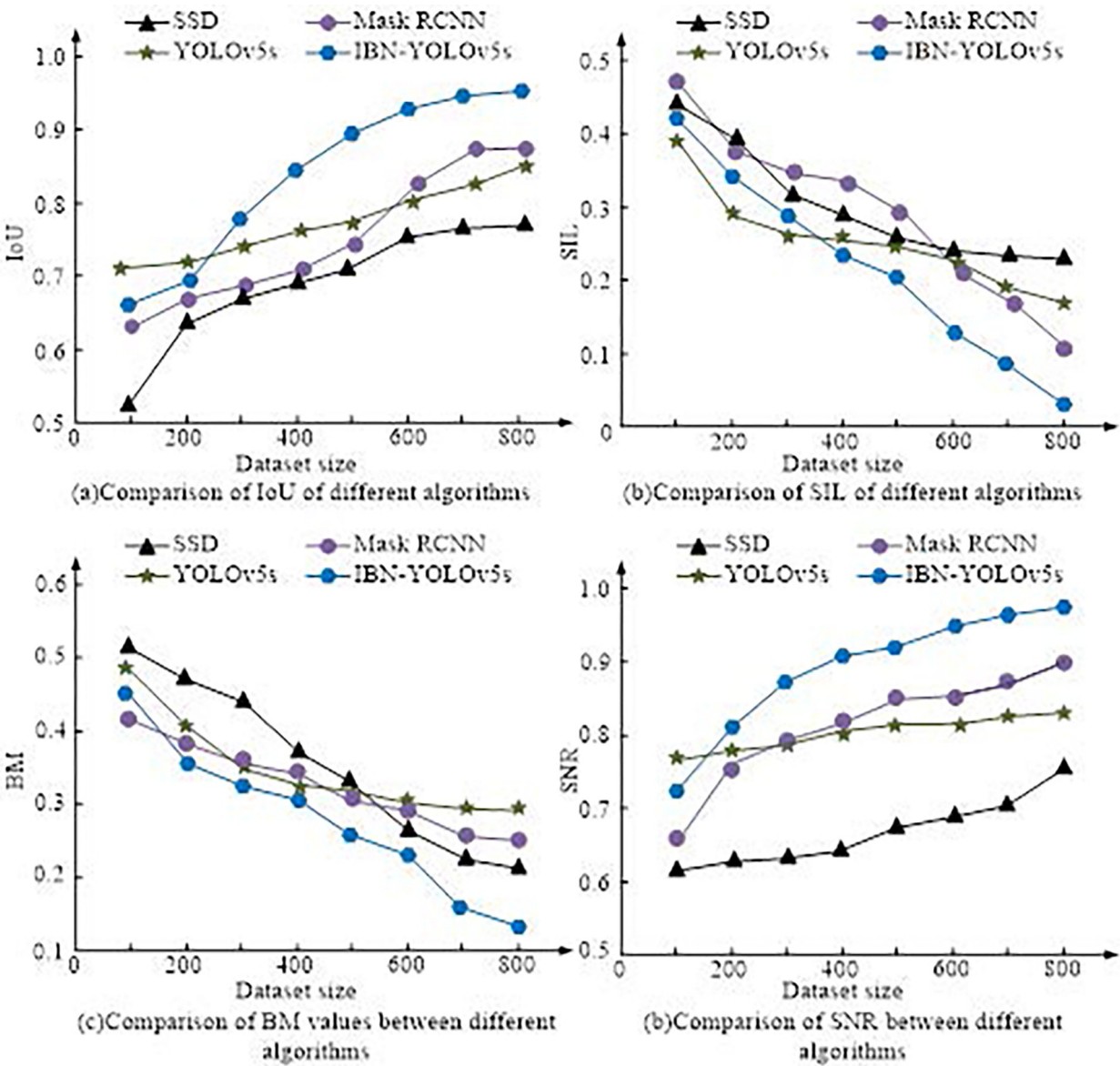

**Fig 8. Performance comparison of various algorithms in training set.**

BM of each algorithm decreased. When the size of the training set was 800, the BM values of SSD, Mask RCNN, YOLOv5s, and IBN-YOLOv5s were 0.23, 0.27, 0.30, and 0.13. In Fig 8B, as the size of the training set increased, the SNR of the images processed by each algorithm showed an increasing trend. When the size of the training set was 800, the SNR values of SSD, Mask RCNN, YOLOv5s, and IBN-YOLOv5s were 0.73, 0.90, 0.82, and 0.98, respectively. Based on the above four indicators, IBN-YOLOv5s has better image processing performance compared to other algorithms. Subsequently, a comparative experiment was conducted on the running time of the four algorithms. The training and testing sets were divided into four datasets with sizes of 200, 400, 600, and 800, respectively. The results are shown in Fig 9.

Fig 9A and 9B showcase the average time consumption of the four algorithms with different data sizes on the training and test sets, respectively. In Fig 9A, IBN-YOLOv5s had the shortest time consumption on each dataset. On the 800 size training dataset, the average time

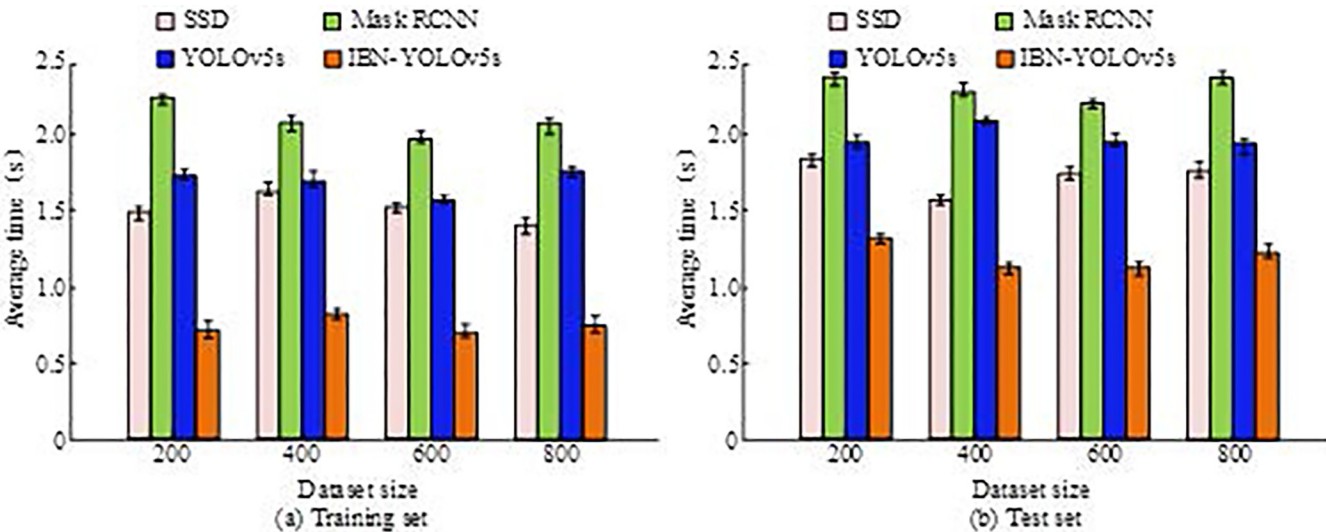

**Fig 9. Running time of four algorithms.**

consumption of SSD, Mask RCNN, YOLOv5s, and IBN-YOLOv5s was 1.4 s, 2.1 s, 1.7 s, and 0.7 s, respectively. In Fig 9B, the IBN-YOLOv5s algorithm also performed well, with an average time consumption of 1.7 s, 2.3 s, 2.0 s, and 1.4 s for SSD, Mask RCNN, YOLOv5s, and IBN-YOLOv5s, respectively. The research results indicated that the IBN-YOLOv5s algorithm had good performance in object recognition.

## 4.2 Experimental testing of basketball robot application integrating IBN-YOLOv5s

On the basis of verifying the precision, IoU, and time consumption of the IBN-YOLOv5s algorithm in the pre-order, the study would further test the practical application performance of basketball robots fused with IBN-YOLOv5s. Firstly, the comparison outcomes of the mAP of the five algorithms are shown in Fig 10. YOLOv5s-CBAN is an object detection algorithm that

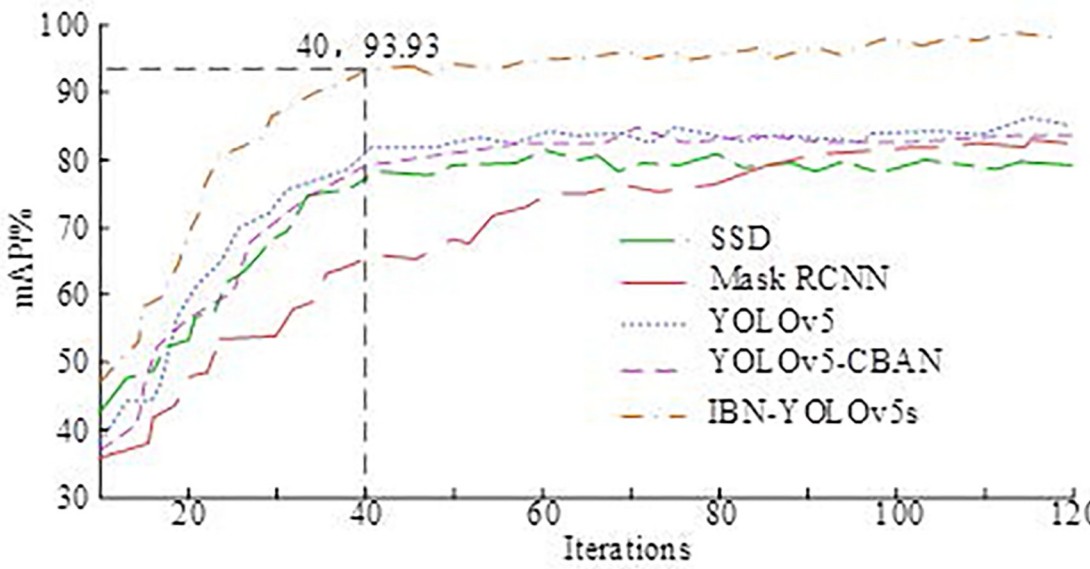

**Fig 10. mAP results of five algorithms.**

**Table 2. Detection performance of four models in three shooting environments.**

| Shooting environment | Test index | Mask RCNN | SSD | YOLOv5s | IBN-YOLOv5s |
|---|---|---|---|---|---|
| Free throw | Identification time /s | 0.19 | 0.16 | 0.09 | 0.02 |
| | Recognition accuracy rate /% | 85.32 | 88.69 | 92.38 | 99.25 |
| Two-point shot | Identification time /s | 0.25 | 0.18 | 0.11 | 0.05 |
| | Recognition accuracy rate /% | 84.05 | 88.37 | 91.05 | 98.89 |
| Three-point shot | Identification time /s | 0.27 | 0.23 | 0.12 | 0.08 |
| | Recognition accuracy rate /% | 82.17 | 87.75 | 90.12 | 97.16 |

combines attention mechanisms (Channel Attention, Bottleneck Attention Network) to help the model focus more on important information during feature extraction, thereby improving detection performance.

In Fig 10, as the amount of iterations increased, the mAP values of all algorithms showed a rapid upward trend in the early stage, and the curve trend gradually flattened after reaching a certain amount of iterations. When the number of iterations was 40, the mAP values of the five algorithms, SSD, Mask RCNN, YOLOv5s, YOLOv5-CBAN, and IBN-YOLOv5s, were 78.23%, 65.01%, 82.16%, 77.21%, and 93.93%, respectively. When the number of iterations was 120, the mAP values of SSD, Mask RCNN, YOLOv5s, YOLOv5-CBAN, and IBN-YOLOv5s algorithms were 79.58%, 82.06%, 86.37%, 84.01%, and 98.43%, respectively. Overall, the mAP value of the IBN-YOLOv5s model always remained at its maximum. To further test the object detection performance of each model in different environments, three different shooting environments were set up, and the comparative tests of the four models are denoted in Table 2.

Table 2 presents the recognition performance of four models under three different shooting environments: free throw, two-point shooting, and three-point shooting. According to Table 2, the recognition time of IBN-YOLOv5s was the shortest in all three shooting environments, with values of 0.02 s, 0.05 s, and 0.08 s, respectively. The accuracy of shooting recognition was the highest, with values of 99.25%, 98.89%, and 97.16%, respectively. To verify the feasibility of the algorithm in identifying different objects, basketball, volleyball, and calibration columns were set up for object recognition experiments. The results are shown in Fig 11.

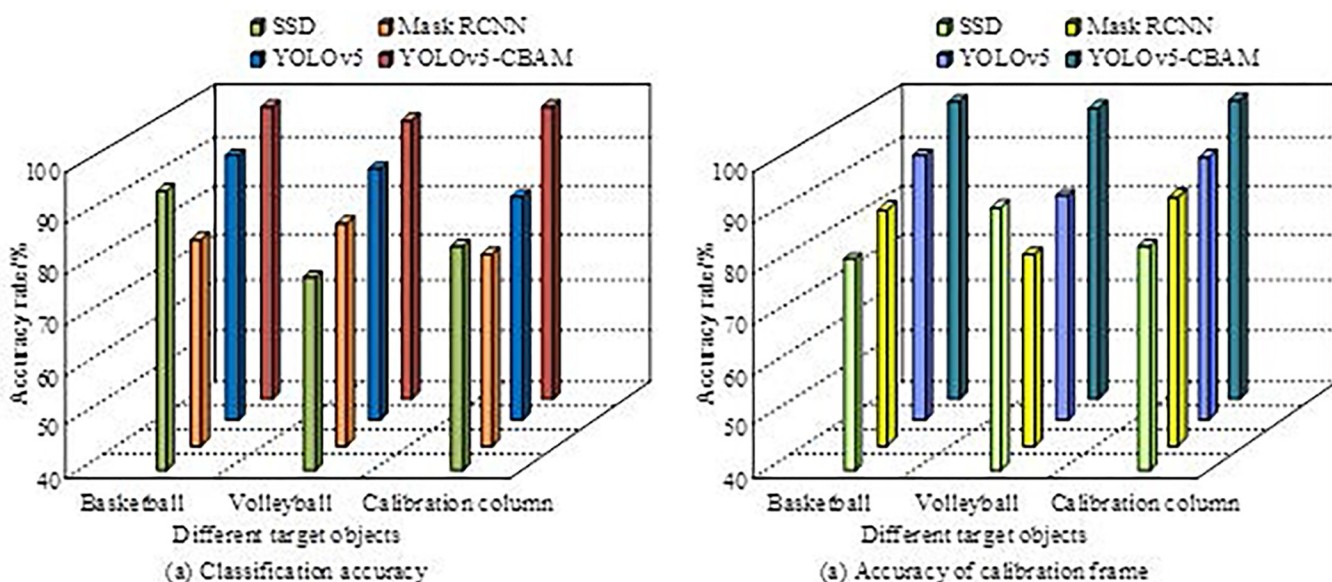

**Fig 11. The classification accuracy and calibration frame accuracy results of four algorithms for different objects.**

Fig 11A and 11B show the classification accuracy and calibration frame accuracy results of the four models, respectively. In Fig 11A, for the three objects of basketball, volleyball, and calibration frame, the classification accuracy of the IBN-YOLOv5s model was 96.49%, 93.87%, and 97.26%, which was better than the other three algorithms. In Fig 11B, the IBN-YOLOv5s model had calibration frame accuracy rates of 97.37%, 95.05%, and 98.73% for basketball, volleyball, and calibration frames, respectively, which were also better than all other comparison models. The research results indicated that the IBN-YOLOv5s model proposed by the research had robust performance and could effectively recognize different objects in basketball robot games.

### 4.3 Discussion

To verify the feasibility of the improved IBN-YOLOv5s algorithm in basketball robot object detection, in performance testing, the F1 values of IBN-YOLOv5s were 97.89% after 150 iterations. The IBN-YOLOv5s model maintained the highest F1 value before and after iterations. When the number of iterations exceeded 150, the F1 value changed gradually and stabilized at over 90%. The reference [10] showed that YOLOv5 had a recognition recall rate of 91.48% and an accuracy rate of 83.83%, mAP 86.75%, F1 value 87.49%. It can be seen that the research algorithm has good performance. In the IoU ratio of processed images, as the number of training sets increased, the IoU values of processed images showed an increasing trend. When the training set size was 800, the IoU values of IBN-YOLOv5s were 0.96. Reference [13] showed that the detection model had an IoU of 97.8% and an accuracy of 89.6%, indicating the advantage of the proposed method in terms of IoU ratio. As the size of the training set increased, the signal-to-noise ratio of the image processed by the algorithm showed an increasing trend. When the size of the training set was 800, the SNR value of IBN-YOLOv5s was 0.98. The improved YOLOv5s recognition proposed in reference [11] had a recall rate of 91.48% and an accuracy rate of 83.83%, mAP 86.75%, F1 value 87.49%. It can be seen that the proposed IBN-YOLOv5s algorithm has better image processing performance compared to other algorithms. In the testing of the basketball robot system integrating IBN-YOLOv5s, the mAP value of the IBN-YOLOv5s algorithm was 98.43% when the number of iterations was 120. The improved algorithm proposed in reference [7] improved the mAP of the YOLOv5 model by 4.3% and 25.8% compared to YOLOv5s and YOLOv4, respectively. It can be seen that the algorithm has better accuracy and precision compared to other algorithms. The results indicate that the IBN-YOLOv5s model proposed by the research has robust performance and can effectively recognize different target objects in basketball robot competitions. Research is not only an exploration of the application of YOLOv5s algorithm in basketball robot games, but also the discovery of new technological paths and solutions through continuous optimization and improvement of the algorithm, promoting the technological innovation and development of deep learning technology in the field of sports robots, which is of great significance for promoting the progress of the entire sports industry.

### 5. Conclusion

To improve the recognition performance of basketball robots in robot object detection competitions, a basketball robot object detection system was designed by combining ROS and IBN-YOLOv5s. In the application layer, the system introduced BN and IN modules into YOLOv5s, proposed an improved IBN-YOLOv5s algorithm, and integrated it with laser detection as the system object detection algorithm, aiming to improve the object detection performance of basketball robots. On the PR curve, the areas enclosed by the SSD, Mask RCNN, YOLOv5s, and IBN-YOLOv5s curves with the coordinate axis were 0.82, 0.69, 0.87, and 0.95,

respectively. When the amount of iterations was 150, the F1 value of IBN-YOLOv5s was 97.89%. Secondly, in the tests of IoU, SIL, BM, and SNR, the IoU of IBN-YOLOv5s was 0.96, the SIL value was 0.03, the BM value was 0.13, and the SNR value was 0.98, showing the best performance among all models. In terms of time consumption, the average time consumption of IBN-YOLOv5s on the training and testing sets was 0.7s and 1.4s, respectively. In the confusion matrix, there were only a few false positives, resulting in excellent recognition performance. In the recognition of basketball, volleyball, and calibration frames, the average classification accuracy of the IBN-YOLOv5s model was 95.87%, and the average calibration frame accuracy was 97.05%. The comprehensive performance of the improved algorithm raised in this study was excellent, but the IBN-YOLOv5s used in the study was a fixed parameter network structure, which will reduce the adaptability of the model. In future research, dynamic parameters can be selected to improve the structure of the network model for further testing.

## Author Contributions

**Conceptualization:** Jirong Zeng.

**Data curation:** Jirong Zeng, Jingjing Fu.

**Formal analysis:** Jirong Zeng, Jingjing Fu.

**Investigation:** Jirong Zeng.

**Methodology:** Jirong Zeng, Jingjing Fu.

**Validation:** Jirong Zeng, Jingjing Fu.

**Visualization:** Jirong Zeng.

**Writing – original draft:** Jirong Zeng, Jingjing Fu.

**Writing – review & editing:** Jirong Zeng, Jingjing Fu.

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
