## [Decision Letter · Decision Letter 0]

9 Aug 2024

PONE-D-24-29367Basketball Robot Object Detection and Distance Measurement Based on ROS and IBN-YOLOv5s AlgorithmsPLOS ONE

Dear Dr. Zeng,

Thank you for submitting your manuscript to PLOS ONE. After careful consideration, we feel that it has merit but does not fully meet PLOS ONE’s publication criteria as it currently stands. Therefore, we invite you to submit a revised version of the manuscript that addresses the points raised during the review process.

We look forward to receiving your revised manuscript.

Kind regards,

Burak Erkayman

Academic Editor

PLOS ONE

Journal Requirements:

Reviewers' comments:

Reviewer's Responses to Questions

**Comments to the Author**

1. Is the manuscript technically sound, and do the data support the conclusions?

Reviewer #1: Partly

Reviewer #2: Yes

2. Has the statistical analysis been performed appropriately and rigorously? 

Reviewer #1: Yes

Reviewer #2: Yes

3. Have the authors made all data underlying the findings in their manuscript fully available?

Reviewer #1: Yes

Reviewer #2: Yes

4. Is the manuscript presented in an intelligible fashion and written in standard English?

Reviewer #1: Yes

Reviewer #2: Yes

5. Review Comments to the Author

**Reviewer #1: **• There are a considerable number of typographical errors that need to be addressed.

• The introduction doesn’t have to be in a single paragraph. I highly recommend dividing it into three paragraphs, as it is currently too long. One of these could focus on the research structure. Similar issues exist throughout the manuscript and should be addressed.

• In the related work section, there are some irrelevant studies that should be removed, and more relevant studies should be included.

• The description of the deployed training facility was not well-explained, especially regarding which GPU was used, which is crucial information.

• It would be beneficial to specify which deep learning library was used.

• A maximum iteration number of 150 seems quite low. Additionally, the batch size used

during training should be specified.

• The explanation of the model was insufficient and needs improvement.

• YOLO has many versions, for instance, YOLOv10, but the author used YOLOv5. It should be explained why YOLOv5 was selected, if possible.

**Reviewer #2: **The authors sought to enhance the performance of basketball robots by advancing object detection capabilities. The study developed an advanced detection system that combines the robot operating system with a fusion algorithm integrating v5s and laser detection technologies. Notable enhancements included the incorporation of instance-batch normalization modules to improve the model's generalization. Although the study presents promising results, several recommendations are offered to improve the manuscript.

Abstract

-The abstract would benefit from a clearer articulation of the need for object detection in basketball robots. It should explicitly outline the motivation behind this research, including the specific challenges addressed and the significance of the study. Enhancing these aspects will provide a more comprehensive understanding of the research question and its relevance.

-Additionally, the manuscript should maintain a consistent written style throughout. For instance, the first percentage is presented as 0.95, while the second is given as 97.89%. It is important to standardize the format for presenting percentages. For example, the statement "The recall curve area and F1 value of the improved algorithm were 0.95 and 97.89%, respectively," should be revised to ensure uniformity in the presentation of numerical data.

Introduction

The following revisions are recommended for this section:

-Citation Requirement:

The sentence "In the Chinese Robot Competition, the basketball robot project has become an important event" requires a citation to substantiate the claim. Please include an appropriate reference to support this statement.

-Sentence Structure:

Some sentences should be divided into more digestible parts and revised for a more formal academic tone. For example, consider revising the following sentence:

"At present, traditional object detection algorithms such as binocular vision algorithms, deep convolutional neural networks, and support vector machines, although they exhibit high recognition accuracy, suffer from slow detection speeds and limited learning and generalization abilities [5]."

-Terminology:

Instead of using informal terms like "Part 1" and "Part 2," the manuscript should use more formal terminology such as "Section 1," "Section 2," etc., to enhance the academic quality of the document.

-Additionally, the contributions of the study should be highligted in this section.

Related Work

-Please ensure careful attention to typographical errors, such as the possible confusion between "ROC" and "ROS."

-Additionally, algorithms discussed in the literature should be systematically presented in a table. The table should include columns for features such as "Authors, Year," "Dataset," "Method," "Performance Metrics," and "Results." This will facilitate a clearer comparison and review of the proposed methods. Comparisons should be conducted using the same dataset to ensure consistency and validity. If a comparable public dataset is unavailable, it is essential to provide detailed information about the dataset used for each literature study, including its characteristics, in a designated column.

Method

-The method is described with clarity; however, a question arises regarding the choice of algorithm. Specifically, why did the authors select the YoloV5s architecture? Given that there are more recent versions of the Yolo algorithm, such as YoloV7 and YoloV8, it would be beneficial to explore whether the authors considered or combined Instance Batch Normalization (IBN) with these newer versions. A comparative analysis of the results from these subsequent versions would provide valuable insights and enhance the discussion.

-Minor correction required: In Figure 9, the color representing the proposed IBN-Yolov5s appears orange rather than yellow. Additionally, the color used for Yolov5s is intended to be purple, but this color is not discernible in the figure. Please adjust the color scheme to accurately reflect the intended distinctions.

-In Figure 10, the authors reference the YOLOv5-CBAM algorithm. However, this algorithm is introduced for the first time in Section 4.2. It would be beneficial to provide an earlier mention or detailed description of this algorithm within the manuscript to ensure clarity and coherence. Furthermore, there is no comparison of the IBN-YOLOv5s algorithm with the other algorithms discussed in the preceding sections; only YOLOv5-CBAM is compared. This discrepancy suggests a possible oversight. Please review and ensure that comparisons with all relevant algorithms are included and accurately represented.

Discussion

-A discussion section should be added to the article to provide a thorough analysis and interpretation of the results, contextualize the findings within the broader field, and address any implications or limitations of the study.

6. PLOS authors have the option to publish the peer review history of their article (what does this mean?). If published, this will include your full peer review and any attached files.

Reviewer #1: No

Reviewer #2: **Yes: **Isil KARABEY AKSAKALLI

---

## [Author Response · Author response to Decision Letter 0]

20 Aug 2024

I upload my response to specific reviewer and editor comments with “revised manuscript”.

---

## [Decision Letter · Decision Letter 1]

2 Sep 2024

Basketball Robot Object Detection and Distance Measurement Based on ROS and IBN-YOLOv5s Algorithms

PONE-D-24-29367R1

Dear Dr. Zeng,

We’re pleased to inform you that your manuscript has been judged scientifically suitable for publication and will be formally accepted for publication once it meets all outstanding technical requirements.

Kind regards,

Burak Erkayman

Academic Editor

PLOS ONE

Additional Editor Comments (optional):

Reviewers' comments:

Reviewer's Responses to Questions

**Comments to the Author**

1. If the authors have adequately addressed your comments raised in a previous round of review and you feel that this manuscript is now acceptable for publication, you may indicate that here to bypass the “Comments to the Author” section, enter your conflict of interest statement in the “Confidential to Editor” section, and submit your "Accept" recommendation.

Reviewer #1: All comments have been addressed

Reviewer #2: All comments have been addressed

2. Is the manuscript technically sound, and do the data support the conclusions?

Reviewer #1: Yes

Reviewer #2: Yes

3. Has the statistical analysis been performed appropriately and rigorously? 

Reviewer #1: Yes

Reviewer #2: Yes

4. Have the authors made all data underlying the findings in their manuscript fully available?

Reviewer #1: Yes

Reviewer #2: Yes

5. Is the manuscript presented in an intelligible fashion and written in standard English?

Reviewer #1: Yes

Reviewer #2: Yes

6. Review Comments to the Author

Reviewer #1: The authors have adequately addressed my comments raised in a previous round of review and I feel that this manuscript is now acceptable for publication.

Reviewer #2: Thank you for applying my recommendations rigirously. I think the manuscript is ready for the publication in this way.

7. PLOS authors have the option to publish the peer review history of their article (what does this mean?). If published, this will include your full peer review and any attached files.

Reviewer #1: **Yes: **

Reviewer #2: **Yes: **

---

## [Editor Report · Acceptance letter]

9 Sep 2024

PONE-D-24-29367R1 

PLOS ONE

Dear Dr. Zeng, 

I'm pleased to inform you that your manuscript has been deemed suitable for publication in PLOS ONE. Congratulations! Your manuscript is now being handed over to our production team.

Kind regards, 

on behalf of

Dr. Burak Erkayman 

Academic Editor

PLOS ONE